# Recommendations for the Development of Family-Based Interventions Aiming to Prevent Unhealthy Changes in Energy Balance-Related Behavior during the Transition to Parenthood: A Focus Group Study

**DOI:** 10.3390/nu14112346

**Published:** 2022-06-04

**Authors:** Vickà Versele, Benedicte Deforche, Dirk Aerenhouts, Peter Clarys, Roland Devlieger, Annick Bogaerts, Christoph Liel, Johanna Löchner, Jörg Wolstein, Mireille van Poppel, Tom Deliens

**Affiliations:** 1Department of Movement and Sport Sciences, Faculty of Physical Education and Physiotherapy, Vrije Universiteit Brussel, 1050 Brussels, Belgium; benedicte.deforche@ugent.be (B.D.); dirk.aerenhouts@vub.be (D.A.); peter.clarys@vub.be (P.C.); tom.deliens@vub.be (T.D.); 2Department of Development and Regeneration, Faculty of Medicine, KU Leuven, 3000 Leuven, Belgium; roland.devlieger@uzleuven.be (R.D.); annick.bogaerts@kuleuven.be (A.B.); 3Department of Public Health and Primary Care, Faculty of Medicine and Health Science, Ghent University, 9000 Ghent, Belgium; 4Obstetrics and Gynecology, University Hospitals KU Leuven, 3000 Leuven, Belgium; 5Department of Obstetrics, Gynecology and Fertility, GZA Campus Wilrijk, 2610 Wilrijk, Belgium; 6Centre for Research and Innovation in Care (CRIC), Faculty of Medicine and Health Sciences, University of Antwerp, 2000 Antwerp, Belgium; 7Faculty of Health, University of Plymouth, Plymouth PL4 8AA, UK; 8National Centre for Early Prevention, German Youth Institute, Department of Family and Family Policies, 81541 Munich, Germany; liel@dji.de (C.L.); johanna.loechner@med.uni-tuebingen.de (J.L.); 9Department of Psychology, University of Bamberg, 96047 Bamberg, Germany; joerg.wolstein@uni-bamberg.de; 10Institute of Human Movement Science, Sport and Health, University of Graz, 8010 Graz, Austria; mireille.van-poppel@uni-graz.at

**Keywords:** family-based, intervention strategy, needs assessment, nutrition, physical activity, sedentary behavior, pregnancy, postpartum

## Abstract

Feasible interventions addressing unhealthy changes in energy balance-related behavior (EBRB) during pregnancy and early postpartum are needed. This study identified the needs and wishes of expecting and first-time parents concerning EBRB interventions during the transition to parenthood. Thirteen focus group discussions (n = 74) were conducted. Couples provided information about whether an intervention targeting unhealthy EBRB changes during pregnancy and postpartum would be acceptable, how such an intervention should look like, and in which way and during which period they needed support. Guided by the TiDIER checklist, all quotes were divided into five main categories (i.e., ‘what’, ‘how’, ‘when and how much’, ‘where’, ‘for and from whom’). Interventions should aim for changes at the individual, social, environmental and policy levels. The accessibility and approach (indirect or face-to-face) together with communicational aspects should be taken into account. A focus should go to delivering reliable and personalized information and improving self-regulation skills. Interventions should be couple- or family-based. Authorities, healthcare professionals, the partner and peers are important sources for intervention delivery and support. In the prevention of unhealthy EBRB changes around childbirth, the involvement of both parents is needed, while health care professionals play an important role in providing personalized advice.

## 1. Introduction

Becoming a mother is associated with greater increases in body mass index (BMI) compared to women who remain without children [1]. Gestational weight gain (GWG) outside the recommended Institute of Medicine (IOM) ranges is a risk factor for short- and long-term postpartum weight retention [2,3]. This might contribute to the development of overweight and obesity later in life, and thus, excessive GWG and weight retention during the postpartum period should be prevented [2,4]. Additionally, fatherhood is associated with increases in BMI trajectories [5]. Even though men are at risk for unhealthy weight and lifestyle development during this life phase [6], (expecting) fathers are scarcely included in research and prevention programs focusing on excessive weight gain during the pregnancy and postpartum period [1,7]. 

Excessive GWG and paternal weight gain can be attributed to unhealthy changes in energy balance-related behaviors (EBRB), such as dietary intake, physical activity (PA) and sedentary behavior (SB) [1,8,9]. Socio-ecological models of health behavior, which explain health behavior by the interplay between the different levels of influence, i.e., individual, social, physical environment and policy levels, can be used to understand these changes [10,11]. Determinants of changes in EBRB during the transition to parenthood were indeed determined at an individual (biological factors, e.g., fatigue; psychological factors, e.g., parenthood perceptions and barriers; and situational factors, e.g., time constraints), interpersonal (e.g., partner support), environmental (e.g., home food availability) and policy level (e.g., parental leave) for both (expecting) mothers and fathers [12,13]. During the reproductive phase, women and men are acceptive of lifestyle interventions aiming to improve health behaviors, with the health of the unborn child as the main motivation [14,15,16]. When developing interventions aimed at preventing excessive GWG and paternal weight gain, it is thus important to target the above-described determinants which have an influence on health behavior. Even though the transition to parenthood can be seen as a window of opportunities to shape healthy lifestyles, interventions during this period, targeting dietary habits, PA and SB, or a combination of these three, have not always been shown to be effective in reducing GWG or PPWR. Even when significant effects of interventions were found, effect sizes seem very small [8,17,18]. In order to develop effective prevention strategies for excessive weight gain during pregnancy and weight retention postpartum, it is important to gain insight into the needs of the target population.

Involving the target population in the development is highly relevant to identifying how, what, where, when, for and from whom program objectives have to be defined and delivered. This will increase the adherence and effectiveness of interventions. The Intervention Mapping (IM) protocol is a widely used framework for the development of theory- and evidence-based behavior change programs [19,20]. One perspective of IM is to involve the target population during the process of intervention development. A clear understanding of the needs and desires of (expecting) parents could help planners in developing effective and feasible interventions aiming to improve EBRBs. Given that qualitative methods are considered particularly useful for understanding the process behind behavior change rather than outcomes, the use of this research approach can offer insights into key elements to explore the needs for interventions in the prevention of unhealthy weight gain and EBRB [21,22,23]. Therefore, the aim of this study was to identify the needs and wishes of expecting and first-time mothers and fathers to prevent unhealthy changes in EBRB during pregnancy and early postpartum.

## 2. Materials and Methods

### 2.1. Design and Participants

In this qualitative study, focus group discussions were used for data collection. Focus groups allow providing in-depth insights on a specific topic through dynamic group interactions. It is a naturalistic (i.e., close to an everyday conversation) approach during which participants—led by a moderator—are encouraged to explore and clarify their perspectives [24,25]. This approach allows us to gain insight from the target population (i.e., expectant and first-time parents) into their specific needs and barriers regarding interventions targeting EBRB during the pregnancy and postpartum period. Participants were recruited using a snowball sampling strategy, i.e., a sample of individuals from the target population was recruited through the use of digital flyers and newsletters of organizations with a target audience of new parents or parents-to-be. Interested volunteers could leave their contact details and were afterwards contacted by the researcher through the mail, telephone or text message. Recruited participants were then asked to pass on names and contact details from other expectant/first-time parents from their social network (after asking permission of the individuals concerned). These contacts were then contacted by the researchers and recruited if they were interested and eligible [26]. For the first set of focus groups (i.e., focusing on the pregnancy period), women and men expecting their first child or who recently entered parenthood (no longer than three months) were included. For the second set of focus groups (i.e., focusing on the postpartum period), first-time parents with a child of three to twelve months were included. Participants were recruited individually (e.g., when their partner did not want to or could not participate) or as a couple. The focus group discussions were organized in a homogeneous (women or men only) and heterogeneous (mixed sex) set-up to ensure diversity of opinions and interactions within and across focus groups. The aim was to recruit between six and eight participants per focus group. For each focus group, an over-recruitment of one or two participants was pursued in case of ‘no-shows’ [27,28]. As sample size can never be pre-determined in qualitative research, focus groups were held until saturation of information was reached [27,28,29]. The study was approved by the Medical Ethics Committee of the University Hospital (Vrije Universiteit Brussel, Brussels, Belgium) (B.U.N. 143201835875). The trial was conducted in compliance with the principles of the Declaration of Helsinki (current version), the principles of Good Clinical Practice (GCP) and in accordance with all applicable regulatory requirements. Participants signed an informed consent prior to participation. 

### 2.2. Procedure

All focus groups were organized in the evening between March and June 2019, on a date and location convenient for the participants (i.e., in a private meeting room or a meeting room of a hospital or healthcare practice spread across five provinces in Flanders (Belgium)). For each focus group, a time of 90 to 120 min was foreseen. This included time for explaining the purpose, signing the informed consent, and answering a short questionnaire, including socio-demographics (i.e., age, ethnicity and level of education: no degree, lower, secondary, college and university education), perceived health (“How would you currently describe your health status in general?” on a 5-point scale), perceived diet quality (“To what extent do you feel you are eating healthy?” on a 5-point scale), PA level (“On how many days during the last week were you physically active for at least 30 min?”), body weight (self-reported, in kg) and height (self-reported, in cm). Prior to the discussion, the meaning of PA and SB in each context (leisure time, work, transport and household) was explained. All focus groups were moderated by the same moderator (VV) and assisted by an observer, who took notes during the discussion and made sure the moderator did not overlook any participants who tried to add comments. These notes were also used for preliminary content analysis, which was performed immediately after each focus group to decide when the point of saturation of information was reached [24]. Each participant took part in only one focus group. Sandwiches and drinks were provided during the focus groups. Afterwards, participants were offered an incentive, i.e., a body composition analysis using a portable bio-electrical impedance analysis device (TANITA MC780SMA).

### 2.3. Question Guide

In a first part of the focus groups, participants were asked to identify changes in energy balance-related behavior (EBRB) (i.e., eating, PA and SB) and factors influencing changes in their EBRB during the pregnancy or postpartum period. Second, participants were asked to share ideas concerning the development of an intervention strategy aiming to counter unfavorable changes in energy balance-related behavior among couples during pregnancy. For the purpose of the present study, only the questions and findings of the intervention development part will be reported. Findings from the first part can be found elsewhere [12]. All discussions were conducted in Dutch following a semi-structured question guide (see Table 1).

### 2.4. Data Analysis

All focus group discussions were audiotaped with the permission of the participants and transcribed verbatim in Microsoft Word using Windows Media Player. The final data analysis was performed after completing all the focus groups. A combined inductive and deductive content analysis approach was used. In the first step, all quotes derived from each question were examined for recurrent instances. Recurrent quotes from the different questions were systematically identified across the data set and grouped together by means of an open coding system. Secondly, a deductive coding strategy was used to derive the main themes from the data with the TiDIER checklist as a guideline [30]. Similar codes were grouped together into more general concepts and five main categories (‘what’, ‘how’, ‘when and how much’, ‘where’, ‘for and from whom’) [30]. The data was then further divided into subcategories. Data from the ‘what’ category were inductively subdivided in three categories, namely ‘intervention aims’, ‘intervention characteristics’, and ‘content and materials’. A next deductive approach was used to categorize the results from the ‘intervention aims’ subcategory into four socio-ecological levels (individual, interpersonal, environmental, and policy), which was based on the socio-ecological model of health behavior and the results of the focus groups focusing on determinants of changes in EBRB [11,12,13]. A final deductive approach was used to categorize the data at the individual level. This was based on the behavior change technique taxonomy developed by Michie and colleagues [31]. Data analysis was done by one researcher (VV) and performed using NVivo9 software [24]. Doubts or disagreements were discussed with another experienced researcher in qualitative data analysis within the same topic (TD) until a consensus was reached. Data obtained from the questionnaires were analyzed using SPSS 27.0 to provide descriptive statistics of the focus group sample. An overview of the stepwise data analysis is shown in Figure 1.

## 3. Results

### 3.1. Study Sample Characteristics

A total of 13 focus group discussions were held, with 7 focusing on the pregnancy period and 6 focusing on the postpartum period. Of these 13 focus groups, 2 included only women, 2 included only men, and 9 were mixed samples. Between 3 and 8 participants took part per focus group, resulting in a total sample of 74 (36 male, 38 female) participants. Sample characteristics are described in Table 2. 

### 3.2. Intervention Development

Following the principles of the TiDIER checklist complemented with principles from the socio-ecological model of health behavior and the behavior change techniques taxonomy [11,30,31], a checklist for the development of interventions targeting EBRB during pregnancy and early postpartum was developed (Appendix A). An overview of the results with all categories to consider when developing intervention strategies aiming to obtain/maintain healthy EBRB during the pregnancy and postpartum period is given in Figure 2. Each category is described and illustrated with quotes below.

#### 3.2.1. What—Content of the Intervention

##### Intervention Aims

The formulated needs regarding intervention aims were described at four socio-ecological levels: (1) individual behavior change, (2) changing social support and social networks, (3) changing environmental conditions and (4) policy and organizational changes. At the individual level, the need is twofold. Firstly, there is a need for shaping knowledge and skills about health-related situational, psychological and biological barriers and how to withstand negative social influences. Secondly, a cognitive-behavioral focus should be on improving self-regulation skills. The mentioned self-regulation skills can be divided into planning and anticipation skills (e.g., time management), increasing self-belief (e.g., focusing on past successes), improving self-control (e.g., the difference between ‘hunger’ and ‘desire’) and self-management (e.g., self-monitoring of body weight), self-care (e.g., finding a balance between own needs and time with the family), self-discipline (e.g., not using pregnancy as an excuse) and self-efficacy (e.g., not being concerned about other’s comments). Especially in the postpartum period, a focus on the ability of self-care should be the main focus. Parents described the need for learning the ability to leave their child with someone else and daring to let go of control. There is a need for support in acknowledging that there is no problem in taking time for oneself and coping with feelings of guilt when leaving the child with someone else. Parents seem to experience difficulties in daring to ask for help and accepting it.
“*If I were a healthcare professional, I would say to young parents who just got a child that they should not feel guilty about bringing their child somewhere else and make time for themselves to cook, to eat well or to exercise. I think there are a lot of people who feel guilty about that and it would be good if healthcare professionals told you to actually do these kind of things.*”(first-time father)

At the interpersonal level, opportunities and improvements in professional support towards healthy behavior were suggested. Any advice needs to be consistent, available and adjusted to the needs of young parents. Changing social norms, improving knowledge and dispelling misperceptions of the wider social environment about nutrition, PA and SB during pregnancy while introducing partner support is recommended during the pregnancy period. During the postpartum period, there is a need for practical social support at home, e.g., help with child care and household tasks. Changes in the physical environment (e.g., home food environment, mobility-friendly environment) and adjusted facilities (e.g., child-friendly sport clubs) were suggested at the environmental level. Finally, economic changes, for example, reimbursed PA coaching or sessions with a dietitian (policy level), and work-related adjustments, such as reduced working hours or reduced work-load in order to have time available to be physically active (organizational level), were suggested during pregnancy. Postpartum, additional changes at the policy and organizational level included changes in existing care, regulations from the government about, e.g., self-care, duration and proportion of parental leave (e.g., possibility to share between mothers and fathers) and changes in child care (e.g., increased opening hours of child care). An overview of suggested intervention aims at an individual, interpersonal, environmental and organizational level, including practical examples, can be found in Appendix A.

##### Intervention Characteristics

Accessibility and getting the right information are very important to expecting and first-time parents. Participants indicated the need to know where to find the right information, without having to make too much effort for it, and for the information to be preferably place- and time-independent. If possible, information should be centralized on one platform that is accessible anywhere and anytime (see also Section 3.2.2)). The approach used to communicate the (messages of the) intervention is the second most important aspect. Participants indicated that it should not be in a ‘preachy’ or forced way and without any obligations or giving them a guilty feeling.
“*I know these topics such as nutrition are important. We already experience quite some pressure. Especially if you are a young parent and you have children, a family … there are so many things you must consider and then you must start working again and there are all kinds of demands, so there is a lot of pressure to do things well … So, you really must facilitate …*”(first-time mother)

Moreover, the quality and framing of information are important. The aim should be to provide evidence-based, clear, reliable and consistent advice. Furthermore, it should not be too much or overwhelming. Participants said it was important how things are called. Reframing of messages should be considered, especially when focusing on PA (e.g., focus on PA instead of sport). For men, a humorous or funny communication style seems to be important. 

A personalized approach, with customized support adjusted to people’s needs and tempo (i.e., both for initializing the intervention when they are ready for change, but also during the intervention itself during their daily activities) should be used. People wanted the possibility to choose between different adjusted options. For the postpartum period, it was suggested to take into account the background or living environment of the people, time efficiency and the financial aspect (i.e., affordable interventions). Service provision was found to be important postpartum. First-time parents wanted interventions incorporated into existing healthcare services without them having to ask for it and bundled with the information about the child which they already receive. 

##### Content and Materials

Participants explained the need for specific, easy and practical advice. This could be done by making use of, e.g., ‘tips and tricks’ or replacement messages (e.g., don’t say “you cannot eat raw vegetables”, say “eat grilled vegetables instead of raw vegetables”). A focus should be on messages about what is still possible during pregnancy or what can be done with the baby. For interventions during the postpartum period, participants said it was important to include time-efficient and time-saving messages and suggested bundling the information together with information regarding the child. There seems to be a need for specific information for (expecting) fathers themselves, but also on how they can support their partners.
“*These apps are always very female oriented, which makes sense, because this is the goal for 90% of the information. But, if it’s not filtered or directed to men, you will have to scroll or search for a very long time, until you finally find what you need. Sometimes it’s fun to read everything, but not all of it is useful to us.*”(first-time father)

Materials and content have to be adapted and should only be delivered during the right period (beginning vs. middle vs. end of pregnancy, early or late postpartum, etc.). It is important that materials visualize the information in an attractive and not outdated way, and they are preferably free. Rewards such as material incentives (e.g., vouchers, discounts, …) were suggested to be included as part of motivation for individual behavior change. 

In terms of nutrition, people asked for clear nutrition guidelines, adjusted recipes or food lists (e.g., pregnancy-proof, breastfeeding-proof), and easy, fast and time-saving menu and recipe inspiration. Focus should go towards a healthy lifestyle instead of diet. Product information (i.e., information about which food products can be eaten safely to, e.g., reduce the risk of toxoplasmosis) is needed during pregnancy, both for women and men. Postpartum, there is a perceived need for practical social support, e.g., from someone who could do groceries or cooking for the parents. In terms of PA, people asked for messages encouraging general PA (e.g., walking). Moreover, participants mentioned a need for adjusted courses or exercises for pregnant women or for exercises that can be done at home, preferably as a couple or with the baby. Postpartum participants suggested more support for active transportation (e.g., promoting going walking with the stroller to the supermarket instead of taking the car). Additionally, the need for professional support (e.g., a personal trainer) to guide them was mentioned.

#### 3.2.2. How—Mode of Delivery and Reachability

Mode of delivery and reachability was divided into two main levels, (1) indirect and (2) face-to-face (Figure 3). Broadcast media, digital media and print media can be used to distribute interventions and reach the population of expecting and first-time parents. Interventions through digital media were often discussed and seemed very popular in this population, as it is a low threshold for many (expecting) parents to get access to what they need through digital media. In the postpartum period, distribution or publicity through places where the target population is often found were also suggested (e.g., waiting rooms, sport clubs, childcare). In addition, support at an individual level was recommended by the participants. This could be done through indirect contact (e.g., online channels such as WhatsApp), which seems to be important in the postpartum period when young time parents feel they have little time, as well as face-to-face contact. Personal recommendations of existing interventions (e.g., application-based) from healthcare providers or peers were seen as an important way to reach this population of expecting and first-time parents. Finally, interventions were suggested at a group level, bringing and targeting parents together (i.e., linked with peer support). An intervention modus that is centralized in one platform and combines both an indirect and face-to-face mode of delivery is recommended:
“*The gynecologist should tell you ‘this is a good app, and not those other ones or those websites that you are consulting’.*”(first-time father)

#### 3.2.3. Where—Intervention Deployment

There is a need for interventions that can be delivered at home (e.g., exercises to do at home, personal support at home), in sport clubs (e.g., adjusted courses for pregnant women, couples, parents with their children) or at restaurants and supermarkets (e.g., product information). Participants also indicated the need for interventions at work (e.g., adjusted work, shorter working days) or through online channels (e.g., personal support through mail/WhatsApp).
“*The dietitian I’ve been seeing mostly guided me online. So I didn’t have to go there. She focused on people who are busy, and it was indeed difficult for me to go there with him [the baby], so she has guided me completely via WhatsApp and e-mail. (…) So yes, this guidance fitted very well with my living environment at that time.*”(first-time mother)

Finally, interventions should be incorporated into existing care (e.g., parental life coach at child care organizations).

#### 3.2.4. When and How Much—Intensity and Duration

Intervention programs should be delivered on a regular basis, starting during pregnancy with postpartum follow-up.
“*We should get better guidance during pregnancy, e.g., now you only get a flyer with tips on how to eat in a varied way. There are so much more possibilities than that. If this topic already gets more attention during pregnancy, in a good, positive and constructive way, it will be easier postpartum to continue with these advices, because you have already learned these skills to eat in a healthy and balanced way.*”(first-time mother)

It is important to take into account personalized timing, and postpartum delivery of the information could be synchronized with advice for the child. Participants indicated that if an intervention started pre-pregnancy, they could have already developed good habits during pregnancy. During each trimester of pregnancy, there is a specific need for practical and social support. However, the right timing for delivering the information is important (e.g., in the beginning, the focus should lie more on nutrition, while later on, it should be more on PA). At the end of the pregnancy period, there is a need for information about the early postpartum period. Postpartum, a distinction should be made between the early postpartum period (first weeks after delivery), the end of maternity leave (around twelve weeks after delivery), and the later postpartum period (starting from five to six months postpartum). A focus on social support is recommended during early postpartum, whereas more information about nutrition and support on PA is needed by the end of maternity leave and during the later postpartum period when people get used to the new family rhythm. 

#### 3.2.5. Target Groups—For and from Whom

##### For Whom

A focus should go on the couple. It seems very important to involve men, both for themselves and for their partners. Moreover, men indicated that they would be more involved when they were targeted as a couple. Postpartum attention should switch from a couple-based focus to a family-based focus, including the child in the intervention. There is a need for the inclusion of the child in sport activities or combining nutritional advice for parents and child. During pregnancy, the social environment of the couple should also be targeted. There is a need for more awareness about nutrition and PA during pregnancy by other people in the living environment of expecting couples. Many couples were confronted with wrong perceptions, comments, feedback, ‘advice’ from colleagues, family and friends:
“*Sensibilization is needed about physical activity during pregnancy, to get rid of these old ideas about physical activity being bad. They just don’t know because they heard from their moms to sit still. Explain them [social environment] what can be done during pregnancy, otherwise this idea will stay.*”(expecting father)


##### From Whom

Authorities, healthcare professionals, partners and peers are important for delivering information and support. Participants indicated that they wanted reliable, well-known organizations to provide interventions and information.
“*I would prefer to receive information from an organization of which you know that it is reliable such as the Child and Family Organization. If they provide information you will trust it.*”(expecting mother)

Postpartum, the government and health insurance companies were mentioned as potential sources. Healthcare professionals are important to support expecting and first-time parents. During pregnancy, some expecting parents indicated that they wanted support from the gynecologist, whereas others thought the gynecologist was already busy with the medical support, and support regarding lifestyle was preferred to come from midwives or general practitioners. Specialists in the field (physiotherapists, PA coaches and dietitians) were mentioned as healthcare professionals from whom support would be accepted. Postpartum, midwives, healthcare specialists in the field or psychologists seem to be the preferred professionals to support first-time parents. There is especially a need for contact with one centralized person integrated into the pre- and postnatal care trajectories. This should be someone with whom expecting or first-time parents have a trusted relationship and who can refer to other healthcare professionals when needed. Therefore, a multidisciplinary approach would be recommended. The female partner is a very important person in order to reach and motivate men.
“*Actually in my case, there is only one person who would be able to encourage me to exercise more, and that is my wife herself*”(expecting father)

Peers such as other pregnant women or expecting fathers and friends in the same period are also important for support. Additionally, role models (e.g., through social media) could be a source of support regarding lifestyle in this period.

## 4. Discussion

The aim of this study was to identify the needs and wishes to prevent unhealthy changes in EBRB during pregnancy and postpartum in a population of expecting and first-time parents. This study provided knowledge that was translated into a unique checklist for researchers and intervention planners. This checklist gives an overview of what (expecting) parents need, how the intervention aims and characteristics should look like, how (expecting) parents can be reached or supported and when, where and from whom an intervention should be delivered. 

Firstly and most importantly, it was clear from our data that couple- and family-based interventions are crucial. Incorporating and targeting both (expecting) mothers and fathers in interventions are needed from the start of pregnancy, while during the postpartum period, activities or health benefits for the child may be included. The involvement and support of partners were highlighted by men especially. Men moreover indicated the need of their partner in order to be reached. Previous research on determinants of changes in EBRB has shown that partners influence each other in both a negative and positive way [12,13]. Encouraging both (expecting) parents to obtain or maintain healthy EBRBs themselves while incorporating aspects of how partners can support each other is a double win. In fact, research has shown that couple-based interventions are more effective than individual-based interventions [32]. Postpartum, couple-based interventions should shift to family-based interventions. Parents have been shown to be important role models for their children [33,34], and the interpersonal determinant ‘role model’ was also indicated as a motivator of changes in EBRBs [12,13]. However, first-time parents also experienced barriers related to self-care, such as perceptions about parenthood and responsibilities parents feel they have, withholding them from being physically active (e.g., taking time to engage in sport activities) [13]. This idea should be reversed, and parents should realise that investing time in a healthy lifestyle is not selfish but for the benefit of the whole family. Moreover, parents seem to experience a need for information and support during specific phases in their child’s development, such as introducing solid foods [35]. A combined intervention focus on EBRB information for parents together with information for the child is recommended. Support is required both from a practical (e.g., sport activities together with the child, ability to take the child along) as well as informative perspective (e.g., information about healthy eating habits for parents and child). 

A second important aspect to consider for the development of interventions in a population of expecting and first-time parents is the accessibility, reachability, regularity and time aspect of the intervention. It is advisable to develop a program on a regular basis with advice adapted to the specific period and barriers related to this period. The biological changes women experience during the pregnancy period are an important barrier to healthy EBRB [12,13]. Postpartum, the new family demands and responsibilities for the child make parents struggle with fatigue whilst they have limited time and energy available for themselves [36]. The influence of the baby, which is linked to practical, situational and time constraints, has also been described in terms of changes in EBRB [12,13]. Biological (e.g., discomfort, fatigue) and situational determinants related to the baby are perceived as very important in explaining changes in eating behavior during the transition to parenthood [37]. They, however, have limited modifiability and are difficult to target. It is, therefore, of uttermost importance that interventions are delivered with a low threshold, adapted to the specific needs during pregnancy/postpartum and with the possibility to modify and personalize the intervention to the individual needs of a couple/family. Accessible group-level interventions (i.e., including peer support) could be delivered through digital media. Research has already shown that maternal education through digital tools and social networking-based health interventions can be effective in changing behavior [38,39]. In addition, it is suggested to add (online) professional support to meet the need for reliable, scientific and personalized support.

Third, it is recommended that interventions target different problems at the same time. There is not only a need for interventions that provide knowledge, guidance and specific advice on EBRB itself, but interventions also need to focus on improving self-regulation skills. Many described determinants of changes in EBRB, such as self-licensing (pregnancy card), sensitivity to others’ opinions, social pressure and discouragement, and parenthood perceptions (e.g., barriers to asking for help), can be linked to lack of self-control, self-management, self-belief, self-discipline, self-efficacy and self-care [12,13]. Including re-attribution and reframing messages (e.g., contradicting that eating healthy is time-consuming or that healthy behavior is beyond people’s own control) are suggested for (expecting) parents who have difficulties with self-licensing. Parents described the need for interventions, including goal setting and self-monitoring techniques. Combined messages and goals focusing on both health outcomes (e.g., healthy delivery, the health of the child) and lifestyle behaviors (i.e., eating behavior and PA) may be helpful. Goal setting and self-monitoring are indeed behavior change techniques that are shown to be effective [40]. Postpartum, behavioral strategies linked with self-regulation were associated with a decreased energy intake [41], but whether these are also effective in increasing PA has not yet been demonstrated [42]. Our results, however, suggest that including behavior change techniques aiming to improve self-regulation might be effective for interventions with a focus on nutrition and PA in expecting and first-time parents. For men, an additional focus on retaining habits, preventing them from changing their habits in an unhealthy direction during pregnancy, should be included. During the postpartum period, as parents seem to experience feelings of guilt when taking time for themselves to engage in PA, addressing self-care aspects is highly recommended.

Finally, there are many different challenges that cannot be countered by one intervention. As shown by the socio-ecological models and the studies on determinants of changes in EBRB [11,12,13], there are social (e.g., social discouragement), environmental (e.g., environmental food/PA availabilities) and governmental (e.g., policy regulations about maternity leave) influences in the living environment of expecting and first-time parents which affect their behavior. Hence, focusing on key individual and interpersonal determinants might not be enough. Concrete policy actions at these levels (e.g., increasing paternal leave, the option to share parental leave between the mother and father, reimbursement of PA and nutrition coaching sessions) are needed. Therefore, the engagement of decision-makers and healthcare authorities at a national and regional level as well as in healthcare settings is equally needed when unhealthy lifestyle and weight gain prevention are pursued. 

### Strengths and Limitations

The use of focus group discussions is the first strength as it provided in-depth, bottom-up insights and explored the needs and perspectives of the target population. This can be considered the first step towards a user-involvement strategy for the development of future interventions. Second, participants were recruited from five provinces in Belgium (i.e., Brussels Capital Region and four out of five provinces in Flanders (East Flanders, Antwerp, Flemish Brabant, Limburg)), and thus, diversity in the geographic distribution in the Dutch-speaking community is ensured. Third, the inclusion of fathers(-to-be) and making use of mixed- and same-sex focus groups enabled a variety of opinions and interactions. Although we tried to formulate suggestions based on sex, our qualitative research design does not allow for statistical inferences, and thus splitting up our findings by sex was not possible. All suggestions should thus be considered on a family-based level. Fourth, including participants ranging from the beginning of the first trimester of pregnancy up to one year postpartum ensures great diversity of participants in terms of duration of pregnancy and age of the child and reduced recall bias. Fifth, the developed checklist can instantly be used by other researchers, healthcare providers, or policymakers involved in the development of interventions in pregnancy and postpartum care.

The biggest limitation of our study is the inclusion of a rather homogenous sample of Caucasian, higher-educated, heterosexual couples and physically healthy participants. This is a common limitation in studies investigating health behavior [43]. The homogenous sample might also be explained by the recruitment approach used for this study. Snowball sampling is a commonly used technique in qualitative research and was a convenient way to recruit enough expecting and first-time parents, especially to recruit male participants, which turned out to be difficult. We neither specifically asked the participating parents whether they were single or not, had socioeconomic burdens or experienced psychosocial stress. Therefore, results might not be extrapolated toward vulnerable or single-parent families. Furthermore, specific aspects of Belgian policies, e.g., pregnancy and maternity leave regulations, may, however, make it so that this aspect might not be comparable to countries with other maternity leave regulations. Future research should further investigate how interventions specifically can be tailored for families with lower socioeconomic status and high psychosocial stress.

## 5. Conclusions

This study adds knowledge on the needs, desires and feasibility of intervention strategies aiming to prevent unhealthy changes in EBRB in a population of expecting and first-time parents. The developed checklist can be used as a basis to set the foundation for large-scale intervention development efforts. Focus should go towards the development of couple-based (pregnancy) and family-based (postpartum) interventions with a personalized approach. Reachability, accessibility, regularity and time aspects of the interventions are important to consider, as well as the mode of delivery and the inclusion of (online) professional support. In order to obtain and maintain behavior change, clear nutrition and PA guidelines should be provided, while self-regulation skills should be improved. 

## Figures and Tables

**Figure 1 nutrients-14-02346-f001:**
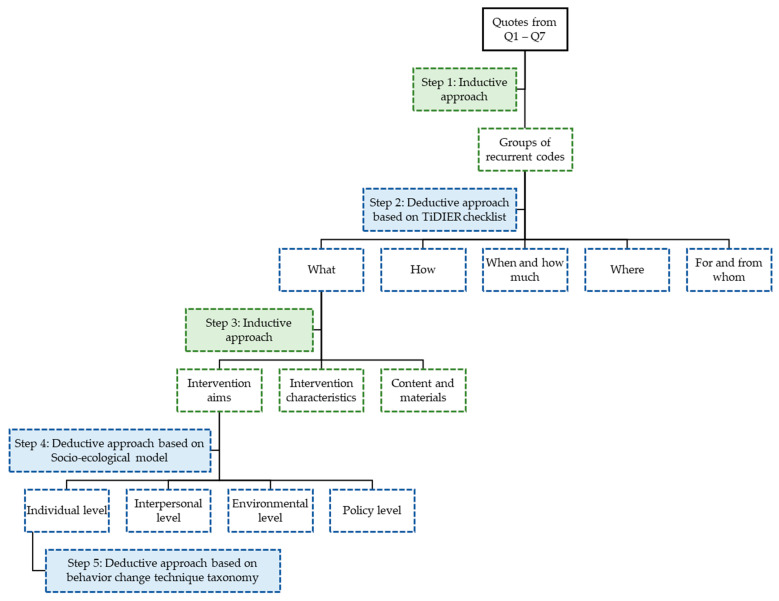
Overview of the inductive—deductive data analysis of the focus group data. Q1–Q7: question 1—question 7. Deductive approach based on: TiDIER checklist [30], socio-ecological model [11] and behavior change technique taxonomy [31].

**Figure 2 nutrients-14-02346-f002:**
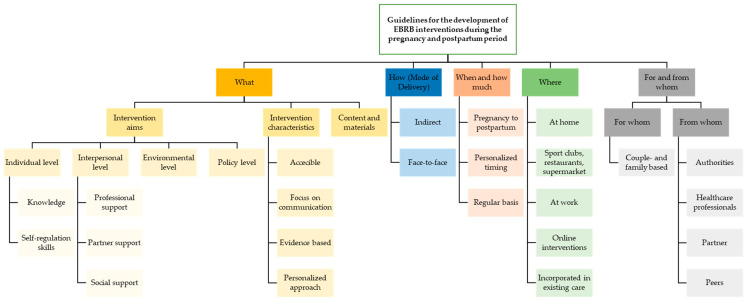
Overview of categories to consider when developing intervention strategies aiming to obtain/maintain healthy energy balance-related behavior (EBRB) during the pregnancy and postpartum period.

**Figure 3 nutrients-14-02346-f003:**
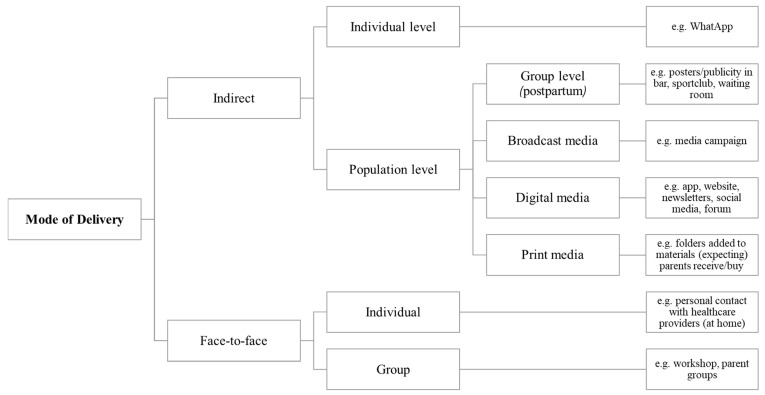
Mode of delivery and reachability of interventions targeting energy balance-related behavior in (expecting) parents.

**Table 1 nutrients-14-02346-t001:** Question guide about intervention development.

Question
1. What could help you to change aspects of your eating behavior, PA and SB during your pregnancy (pregnancy of your partner), or since you became a mother (father)?
2. We will develop an intervention to help couples who are getting (or just got) a first child make healthier choices in terms of eating behavior, PA and SB. What do you think of this?
3. Can you give us advice on what to focus on when promoting healthy eating, adequate PA and limiting SB during pregnancy (during pregnancy of your partner), or after the birth of your child?
4. In which way would you like to be guided and supported to make healthy choices in terms of eating behavior, PA and SB during pregnancy (during the pregnancy of your partner), or after the birth of your child?
5. Through which channels/in which way can we best reach you with such an intervention?
6. In which period (when during pregnancy (in first and second set of focus-groups), shortly after delivery, when your child is a bit older (second set of focus groups)) do you feel the biggest need for support to make healthy choices in terms of eating, PA and SB?
7. If you were a healthcare provider involved in supporting couples expecting or having their first child, name one thing you would do to help couples make healthy choices in terms of eating, PA and SB.

PA: physical activity; SB: sedentary behavior.

**Table 2 nutrients-14-02346-t002:** Characteristics of the participants.

	Focus Groups on Interventions Targeting Changes in EBRB during Pregnancy		Focus Groups on Interventions Targeting Changes in EBRB Up to 1 Year Postpartum	
	Women	Men	Women	Men
Total sample (n)	22	20	16	16
Ethnicity (% of Caucasian)	100	100	100	100
Age (years, mean ± SD)	30.1 ± 2.5	31.6 ± 2.5	30.3 ± 2.0	31.7 ± 3.5
Self-reported pre-pregnancy BMI (kg/m^2^, mean ± SD)	22.7 ± 3.1	24.0 ± 4.5	23.3 ± 4.7	25.0 ± 2.4
Respondents with a higher education (%)	81.8	75.0	93.8	87.5
Perceived health: Respondents reporting to be in good to very good health (%)	100	100	62.6	81.3
Respondents reporting a healthy to totally healthy eating pattern (%)	77.3	80.0	93.8	62.6
Respondents reporting being physically active for at least 30 min/day for 5 days or more during the last 7 days (%)	49.9	45.0	6.3	37.5
% non-smokers (% ex-smokers)	100 (4.5)	100 (40.0)	100 (0.0)	100 (12.5)
Expecting parents (n)	15	14		
Gestational age (weeks, mean ± SD)	28.4 ± 8.1	28.2 ± 8.6		
Parents with child (n) *	7	6	16	16
Age of the newborn (weeks, mean ± SD)	9.6 ± 2.8	9.8 ± 5.2	34.8 ± 14.7	32.6 ± 15.3

* For the focus groups during pregnancy, both expecting parents, as well as parents with a first child less than three months old, participated.

## Data Availability

The audiotapes and transcribed interviews generated and analyzed during the current study are not publicly available due to their containing information that could compromise the privacy of research participants but are available from the corresponding author on reasonable request.

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
