# Peer review of "Recommendations for the Development of Family-Based Interventions Aiming to Prevent Unhealthy Changes in Energy Balance-Related Behavior during the Transition to Parenthood: A Focus Group Study"

_nutrients, 2022, doi:10.3390/nu14112346_

Round 1
Reviewer 1 Report
Methods should be improved since results may be considered arbitrary, most objective methodology would be auspicable, otherwise this limitation should be further addressed.
Author Response
Dear reviewer,
Thank you for your feedback on our manuscript entitled “Recommendations for the development of family-based interventions aiming to prevent unhealthy changes in energy balance related behavior during the transition to parenthood: a focus group study”. Our team of authors has done the best we could to further improve our manuscript. Feedback from the reviewers was sometimes conflicting, e.g., one reviewer had the opinion that the results were clearly presented, whereas the same was questioned by the other reviewer. We have tried to adapt the manuscript where possible while meeting the feedback of the reviewers, and we believe to have improved the manuscript accordingly. All comments were addressed and discussed, and all the changes that have been made are highlighted using track changes in the originally uploaded version of the manuscript.
We look forward to receiving your response.
With kind regards and on behalf of the co-authors,
Vickà Versele

Reviewer 2 Report
The study is quite relevant and so is the study design chosen by the researchers. But there are parts of the paper that are part of a template for how to write papers in each section. So I suggest that instead of having that information you just put information from the study and not from standard documents.
Author Response

(The authors gave the same response as above.)

Reviewer 3 Report
The present study addresses an interesting topic in the lifestyle of first-time parents.
Among the most important observations I would like to highlight:
It is advisable to present the abstract without divisions (remove words such as introduction, methods, etc.) and make it a single text.
The final paragraph of the introduction explaining how the references will be inserted is confusing and is not necessary. The introduction should be limited to the context of the research and end with the justification of the research.
Mention the geographic site where the participants were targeted.
Explain well the method on which the procedure is based. Some phrases such as "master's student" is not appropriate for a scientific article.
For the type of study conducted, the sample size is small.
Table 2 does not report statistical data.
Correct subheading 3.2.1.
The study lacks quantitative data, so robust results and conclusions cannot be drawn.
Some references are very old.
Author Response

(The authors gave the same response as above.)

Round 2
Reviewer 3 Report
The authors were able to correct the observations suggested in the first evaluation, resulting in a manuscript with greater understanding.
They were also able to give greater veracity and scientific rigor to the study.